# The Real-Time Monitoring of the Laser-Induced Functionalization of Transparent Conductive Oxide Films

**DOI:** 10.3390/nano13192706

**Published:** 2023-10-05

**Authors:** Takuya Hosokai, Junichi Nomoto

**Affiliations:** 1National Institute of Advanced Industrial Science and Technology (AIST), Research Institute for Material and Chemical Measurement, National Metrology Institute of Japan (NMIJ), Central 5, 1-1-1 Higashi, Tsukuba 305-8565, Ibaraki, Japan; 2National Institute of Advanced Industrial Science and Technology (AIST), Advanced Manufacturing Research Institute, Central 5, 1-1-1 Higashi, Tsukuba 305-8565, Ibaraki, Japan; nomoto.junichi@aist.go.jp

**Keywords:** transparent conducting oxide films, laser irradiation, photofunctionalization, real-time monitoring, Nd:YAG laser

## Abstract

Laser-induced functionalization using excimer laser irradiation has been widely applied to transparent conductive oxide films. However, exploring suitable irradiation conditions is time-consuming and cost-ineffective as there are numerous routine film fabrication and analytical processes. Thus, we herein explored a real-time monitoring technique of the laser-induced functionalization of transparent conductive oxide films. We developed two types of monitoring apparatus, electrical and optical, and applied them to magnetron-sputtered, Sn-doped In_2_O_3_ films grown on glass substrates and hydrogen-doped In_2_O_3_ films on glass or plastic substrates using a picosecond Nd:YAG pulsed laser. Both techniques could monitor the functionalization from a change in the properties of the films on glass substrates via laser irradiation, but electrical measurement was unsuitable for the plastic samples because of a laser-induced degradation of the underlying plastic substrate, which harmed proper electrical contact. Instead, we proposed that the optical properties in the near-infrared region are more suitable for monitoring. The changes in the optical properties were successfully detected visually in real-time by using an InGaAs near-infrared camera.

## 1. Introduction

In_2_O_3_-based transparent conducting oxide (TCO) films are widely used as transparent electrodes in various opto-electronic devices, such as displays, light-emitting diodes, and solar cells [1,2,3,4]. For instance, solid-phase crystallized hydrogen (H)-doped In_2_O_3_ (IO:H) as well as H and transition metals co-doped In_2_O_3_ (such as Ce and ICO:H) made using the magnetron sputtering technique have been frequently employed for high-efficiency solar cell applications, because those TCO films exhibit a high transparency over a wide range from visible to near-infrared (NIR) [5,6]. However, there is a critical drawback in using flexible substrates, such as polyethylene terephthalate (PET), for these TCO films because of the lowering of the electrical resistivity (*ρ*); *ρ* is inversely proportional to the carrier mobility (*μ*) and carrier concentration (*N*) [7]. The crystallinity of the TCO films is enhanced via annealing, either during or post deposition over 150 °C. On PET, the enhancement of *μ* and/or *N* by controlling defects and crystallization is prohibited by the limit of the thermal annealing temperature (less than 150 °C).

To resolve this issue, the excimer laser irradiation (ELI) technique has been used [8,9,10,11]. Excimer lasers have a high photon energy. For example, KrF (wavelength λ = 248 nm) excimer lasers can achieve a photon energy of 5 eV, which should be sufficient to induce a photothermal reaction. The temperature inside the film exceeds 400 °C, but only during a very short irradiation time (decay over several hundreds of nanoseconds); therefore, thermal damage to the underlayer can be avoided [10,11]. Consequently, we obtained a low sheet resistance (R_s_) of 14.2 Ω/□ (ρ = 2.13 × 10^−4^ Ωcm) stemmed from *μ* of 133 cm^2^/Vs, which is the highest value among the reported flexible TCO films, and a high average optical transmittance in the visible to NIR region [12].

The irradiation conditions of ELI are the repetition rate, power, and the number of shots. To obtain the desired TCO films, suitable laser irradiation conditions must be explored, including the deposition of numerous films and the application of varying laser irradiation conditions to these films. Subsequently, the effects of laser irradiation are verified through various assessments, such as electrical property measurements through Hall effect measurements, X-ray diffraction (XRD), hard X-ray photoemission spectroscopy, and particle analysis using electron microscopes. Such experiments are indispensable for the fabrication of high-performance TCO films; however, these studies incur various costs, such as time, materials, and power. A real-time observation of film properties during ELI should be ideal to reduce the cost of exploring the laser irradiation conditions for enhancing or functionalizing TCO films [13].

Several monitoring techniques of laser-induced crystallization have been developed and applied for semiconducting materials. Previously, Bostangjogo et al. used time-resolved transmission electron microscopy (TEM) and investigated a crystallization process of amorphous silicon (a-Si) and germanium (a-Ge) films on a nanosecond (ns) time scale [14,15,16]. Another group also used time-resolved TEM to characterize the crystallization dynamics of a-Ge films [17,18] and unveiled its details using a multi-frame technique [19]. Optical methods are also utilized for the monitoring of laser-induced crystallization. Lee et al. developed a double laser recrystallization system combined with optical microscopy and determined a lateral solidification velocity of a-Si films [20]. Kuo et al. employed optical transmission and reflection spectroscopy and studied the phase transitions and exclusive crystallization of a-Si films on a ns time scale [21,22,23]. As an alternative to laser irradiation, some investigations to monitor the crystallization of amorphous films during sample heating were conducted using optical spectroscopy, atomic force microscopy, the electrical resistivity measurement method, and in situ XRD [24,25,26,27,28,29,30,31,32,33]. Some works within the literature utilize both pulsed lasers and probe lights in the visible light region because Si and Ge films can absorb light in that wavelength region. However, these lasers or lights are hardly applicable or ineffective for TCO films, owing to their high transparency in the visible region. On the other hand, the use of excimer lasers for the development of a real-time monitoring method has other issues. The excimer laser that is needed to crystallise TCO films via light adsorption is challenging to transport to other facilities, e.g., a TEM facility or synchrotron beamlines to conduct XRD, owing to its large size [34]. Therefore, to facilitate the exploration of the laser irradiation conditions of TCO films, the new development of portable and real-time monitoring techniques that are feasible at the laboratory level is desirable.

Accordingly, we propose real-time monitoring techniques to explore the laser irradiation conditions that are aimed at enhancing the functionality of TCO films. For this development, we employed the fourth-harmonic generation (FHG) of a picosecond Nd:YAG laser instead of an excimer laser, owing to limitations in the setup location. While the energy of FHG (4.66 eV) is lower than that of our excimer laser (5 eV), the significantly higher peak power of our FHG laser, which is 8.6 times greater than that of our excimer laser, led us to anticipate the possibility of achievement of laser-induced functionalization using FHG. We developed electrical and optical measurement methods. We first introduce the results of the development of the electrical method while noting issues with the electrical measurements. To address the issues identified in the electrical measurement techniques, we also developed optical methods. In this regard, we highlight the effectiveness of observing changes in intensity in the NIR wavelength region (1600~1650 nm) derived from variations in the carrier concentration, particularly using an InGaAs camera, as a means of the real-time observation of photofunctionalization. The optical method developed in this study proves to be particularly effective in cost-effectively exploring laser irradiation conditions for TCO films, as it detects the photofunctionalization in large areas and could be adopted to the actual manufacturing process.

## 2. Materials and Methods

The specimens used in this study were radio frequency (RF) (frequency 13.56 MHz), magnetron-sputtered, 120 nm thick indium tin oxide (ITO) films on a quartz glass substrate, and ICO:H films on glass or PET substrates. The films were fabricated without intentionally heating the substrates. For these specimens, the FHG of a picosecond diode-pumped solid-state Nd:YAG pulsed laser (PL2211, Ekspla) was used for the photofunctionalization of the TCO films. The pulse width was ~28 ps, and the beam diameter was ~3 mm (Gaussian beam, 1/e^2^). The irradiation fluence was 0.43–0.45 mJ/pulse, monitored with a thermal sensor (3A, Ophir).

Two types of real-time monitoring apparatus, based on electrical or optical measurements, were developed. The method for monitoring the electrical properties is shown in Figure 1. The samples were placed on a ceramic heater and fixed using screws and metal washers from the top films. An electrometer/high-resistance meter (B2987, Keysight) was used for the voltage supply and current measurements, and a thermocouple (TC) data logger (TC-08, Pico Technology) was employed for temperature measurements. Lead wires were used as electrodes. The electrodes were in direct contact with the sample substrate, and the FHG was irradiated between the electrodes. The terminal spacing of the electrodes was set to approximately 4 mm to avoid irradiating the electrodes using FHG. The TC wires were placed between the electrodes, but near the centre, to avoid the direct irradiation of the wires via FHG. It is noted that no adhesives were used for the contact of both the electrodes and TC wires with the samples to measure the same sample several times by changing the irradiation spots. The FHG, which was transmitted approximately 10% through the ITO samples, was either absorbed or diffusely reflected from the surface of the ceramic heater on the backside of the substrate. The change in resistance after FHG irradiation was measured approximately 10 times at 2.7 ± 0.5 mm between the electrodes using a digital multimeter (CDM-6000, CUSTOM), and the average value was calculated. To identify the effect of FHG irradiation using other measurement methods, raster FHG irradiation was performed using two motorised transition stages (MTS50-Z8, Thorlabs). Subsequently, Hall effect measurements (Resi Test 8300, TOYO Corporation) and out-of-plane *θ/2θ* XRD measurements (SmartLab XRD System, Rigaku Corp.) were performed.

The optical properties were monitored using two methods. Figure 2a,b depict the systems employing a mode-locked supercontinuum source (SC) (SC-Pro-M-40 MHz-PP, YSL Photonics) and a home-made LED ring UV and NIR light, all of which were used as probe lights. To investigate the effect of FHG irradiation, Figure 2a shows the changes in the NIR transmittance, whereas Figure 2b shows the image changes in the UV and NIR diffuse reflectance using two digital cameras. In Figure 2a, the SC light is directed onto the sample surface along the normal direction of the substrate without any focus. A parabolic mirror (MPD169-P01, Thorlabs) and a cold mirror (CLDM-50S, Sigma) were used to observe the transmitted light spectra above 700 nm using a photonic multichannel analyser (C10028-01, Hamamatsu). To obtain two-dimensional images, raster FHG irradiation was performed using motorised transition stages, and the light transmitted through the samples was detected using a biased InGaAs photodiode (DET20C2, Thorlabs) and B2987. For this measurement, the transmitted light was passed through a 1650 nm bandpass filter (BPF). As shown in Figure 2b, a custom-made ring light alternating between 395 nm LED (LED395L, Thorlabs) and 1600 nm LED (LED1600L, Thorlabs) was directed onto the sample to capture diffuse reflectance images using a monochrome CMOS camera (CS165MU, Thorlabs) and an InGaAs NIR camera (XS-USB-FPA-1.7-320-TE0-60 Hz-NTSC-VisNIR, Xenics). The diffuse reflectance of the UV and NIR light was separated using a cold mirror, and spectral separation was further achieved using 390 nm (FB390-10, Thorlabs) and 1600 nm (65794, Edmund Optics) BPFs. Both cameras were simultaneously operated at 10 fps. The FHG and UV/NIR light that were transmitted through the sample were blocked using an aluminium protective screen (TPS5, Thorlabs), installed on the backside of the sample.

In all the measurements, the incidence of FHG in the sample was at angles of less than 12° from the sample normal. All the measurement programs were developed using LabVIEW software. The measurements were conducted at room temperature (296 K).

## 3. Results and Discussion

### 3.1. Electrical Monitoring System: Contact Method

#### 3.1.1. ITO/Quartz Glass

First, we investigated the effect of FHG irradiation on the electrical properties of ITO films. Figure 3a,b show the temporal results of temperature and current of an ITO film grown on a quartz glass substrate using FHG irradiation under a bias voltage of 0.5 V. Under irradiation, both the temperature and current increased immediately, and the temperature rose to 51 °C from an initial value of 26 °C. After the irradiation was stopped, the temperature decreased steeply and returned to the initial temperature. This result was reproducible after a second irradiation. On the other hand, the current also increased but the rate of increase was lower than that of the temperature. Such slow behaviour was also observed after stopping the irradiation; the current at high temperatures remained constant for ~300 s and dropped to a value slightly larger than the initial base current. A larger base current was also observed after the second irradiation. It is supposed that the resistance of the ITO films was reduced through FHG irradiation. However, a large drift in the base current was observed after the two irradiations. For instance, the base current after the first irradiation increased by 2.5%, and further increased by 1.0% owing to the drift. Thus, long-term irradiation can enable a change in the electrical properties of the ITO film but may not be suitable to rigorously assess the improvement in the properties via the present contact method. The large drift might be caused by the latent heat of the sample system, owing to laser absorption.

To clarify the effect of FHG irradiation, short-term irradiation was conducted on a fresh spot of the ITO film. Irradiation for 5 s was repeated five times on the same spot. The temporal results of the temperature and current are shown in Figure 3c,d, respectively. The temperature increased transiently by only approximately 3 °C under each irradiation, showing that the short-term irradiation largely suppressed the increase in the temperature. For the irradiation processes, the current also transiently increased by 3–5% from the initial current and had no delay with respect to the temperature, which differed from the long-term irradiation condition (Figure 3a,b). Remarkably, the base current after the final irradiation maintained a larger value of ~3% compared with the initial value, and no drift to a larger current was confirmed. These results suggest that the resistance of the ITO film decreased upon FHG irradiation. The resistance of the sample across the irradiated spot was measured 20 times using the two-terminal method with a multimeter. The averaged resistance and standard deviation were evaluated to be 275 ± 36 Ω, which was significantly lower than that before the irradiation, that is, 325 ± 24 Ω. It is clear that the resistance of the ITO sample was reduced through FHG irradiation.

The improvement in the electric properties of the ITO films through FHG irradiation was investigated quantitatively using Hall effect measurements and out-of-plane XRD. For these measurements, the FHG was irradiated on a large area (8 × 17 mm^2^) of another ITO film on quartz glass by scanning the irradiation; the sample was moved 1 mm at a time and irradiated for 25 s at each spot. The effect of the irradiation is shown in Figure 4a, where the area shows a lamellar texture by achieving a higher transparency. The resistance was 132 ± 20 Ω in the irradiated area, whereas that in the non-irradiated area was 275 ± 12 Ω. The results of the Hall effect measurements are summarised in Table 1. The electrical resistivity was improved from 5.25 × 10^−4^ Ωcm to 2.73 × 10^−4^ Ωcm, confirming the improvement in the electrical property of the ITO film via FHG irradiation, similar to ELI [35].

An analysis of the results of the Hall effect measurement (Table 1) revealed that the improvement in the electrical resistance of the ITO films was caused by an increase in the carrier concentration by a factor of two and not by Hall mobility. The out-of-plane XRD patterns in Figure 4b show a slight enhancement in the crystallinity of the ITO film. The as-deposited ITO films exhibited 222 diffraction peaks, corresponding to a cubic In_2_O_3_ bixbyite structure, which seemed to split into two peaks at 29.8° and 30.1°. The splitting into two peaks is attributed to the difference in crystallization between the vapour and solid phases that were produced in the deposition process [36]. After FHG irradiation, the intensity of the lower-diffraction-angle reflection decreased, whereas that of the higher-diffraction-angle reflection increased. This indicates the FHG-derived acceleration of film crystallization. However, for the more crystalline ITO films, no change in the Hall mobility was observed (see Table 1). The FHG irradiation activated the Sn dopants via the dissociation of the interstitial oxygen, which formed a neutral cluster with Sn, thus increasing the number of charge carriers [35,37,38]. The decrease in defects is attributed to the transparent optical property of the irradiated area, shown in Figure 4a, as a decrease in the defect states in the band gap of the ITO films within the visible light region also occurred. The carrier concentration increased to an order of magnitude of 21, which suggests that the ionized impurity scattering by the Sn dopants also significantly affected the Hall mobility [39]. Thus, the hypothetical increase in Hall mobility owing to crystallization was compensated by the ionized impurity scattering caused by the increase in carrier concentration.

Finally, we tested the effect of the laser repetition rate on the electrical properties of the ITO films. The ITO films were irradiated with FHG pulses at four different repetition rates of 1 Hz, 10 Hz, 100 Hz, and 1 kHz, while maintaining the total number of irradiated pulses at 20,000. The results are summarised in Table 2. Upon decreasing the rate, the degree of temperature increase (ΔT) was reduced significantly. As the repetition rate was decreased from 1 kHz to 100 Hz, the ΔT became less than 1 °C, and at 1 Hz, no ΔT was detected. In spite of the different ΔTs, the resistance decreased to nearly the same value in all the conditions. These results indicate that the improvement in the electrical properties occurred owing to one pulse irradiation event, and that the degree of the improvement is depended on the number of pulses. While it is known that the electrical properties of ITO films are improved via heating at 200 °C, our results suggest that a photochemical reaction via FHG can contribute to the improvement, which is known for ELI [35].

#### 3.1.2. ICO:H/PET Films

Subsequently, we employed the established electrical monitoring technique for the ICO:H films grown on PET films, which are candidates for flexible electrodes. However, when starting the irradiation, damage to the specimens was easily observed. Figure 5a shows the photographs of the specimens as a function of the number of irradiated pulses. Whilst 100 and 1000 pulses exhibited no change, 2000 pulses induced a circle-like deformation. Furthermore, the deformation increased as the number of pulses increased. At the same laser power condition, we tested the effect on the current measurements and observed poor reproducibility; the current either increased or decreased after the irradiations (see the Appendix A).

As the current was measured via direct contact with the lead wires on the samples, the poor reproducibility was plausibly caused by the deformation of the plastic substrate. Such deformations of the ICO:H/PET films have been reported for ELI [39]. The excimer laser had a thermal effect on the plastic films, and the deformation of the PET films broke the top layer of the ICO:H films. Such deformation was avoided by inserting a SiO_2_ buffer layer between the ICO:H and PET films, and the electrical properties of the ICO:H/PET film were improved [40]. For the ICO:H/SiO_2_/PET samples, we also conducted FHG irradiation and monitored the current (Figure 5b). We obtained an increase in the base current by 7.7% after irradiation for 194 s. However, as shown in Figure 5b, the current suddenly increased transiently when the irradiation was stopped. This was not the case for the ITO/quartz glass samples (Figure 3b). It can be assumed that, during FHG irradiation, the PET films of the ICO:H/SiO_2_/PET samples were also slightly deformed, and unstable electrical contact was induced. Consequently, we conclude that electric monitoring through direct contact can be adopted for glass substrates but is not suitable for PET film substrates, owing to the issue of electrical contact.

### 3.2. Optical Monitoring System: Non-Contact Method

#### 3.2.1. Single-Point Measurement System

The transmission spectrum of the ICO:H/glass substrate is shown in Figure 6a. Through ELI crystallization, the ICO:H film exhibited a sharper change in the transmission peak at around 394 nm, which corresponds to a bandgap of the film, as well as an increase in the transmittance in the NIR region over 1000 nm by approximately 10%. Figure 6b illustrates the time-dependent change in the relative transmission intensity (ΔI) of SC light at a wavelength of 1650 nm before and after FHG irradiation. Here, the ΔI is defined as ΔI = (I_0_ − I)/I_0_ × 100 [%], where I_0_ represents the average intensity before the irradiation. Despite the relatively significant noise of approximately ±1% arising from the stability of the SC light source, the rapid change in ΔI, owing to the FHG irradiation, appeared exponentially, resulting in an average increase of approximately 7.7% (refer to the simulation curve in Figure 6b). Upon ceasing FHG irradiation, although the ΔI momentarily decreased, it remained 4.5% greater than that before irradiation. As such, ΔI changes were not observed with the substrate alone, so it can be concluded that the observed ΔI after the FHG irradiation was due to a change in the optical property of the ICO:H film.

Next, the transmitted SC light intensity at a wavelength of 1650 nm through the ICO:H/glass was detected as a two-dimensional (2D) image using the InGaAs detector. Figure 7a,b show photographs of two samples: a bare glass substrate and ICO:H/glass. These samples were scanned two-dimensionally using motors, and the 2D maps of the transmitted intensity are shown in Figure 7c,d. In the case of the bare substrate (Figure 7c), the areas around the substrate, that is, the air region, exhibited an intense red colour. Based on the intensity in the air region, the transmittance in the substrate area, displayed in orange, was calculated to be 94%. For the ICO:H/glass sample, the transmittance decreased further, as shown in Figure 7d, to approximately 70%, which is similar to the transmittance shown in Figure 6a before laser irradiation. For this sample, FHG was irradiated along the Z-axis direction for only one scan, and a magnified view of the transmittance 2D image is shown in Figure 7e. The colour near the X-axis of the FHG irradiation at approximately 2 mm shifted towards warmer tones in the Z-axis direction, indicating an increase in transmittance. The results for the range of Z = 13–14 mm in Figure 7e, extracted in the X-axis direction, are shown in Figure 7f. The spectrum is observed as a Gaussian function, centred at approximately 2 mm. By employing Gaussian curve-fitting on the result, the FWHM of the region where the intensity increased was observed to be 0.55 mm, and the range determined from the threshold of the Gaussian curve was 1.11 mm. Given that the beam diameter of the FHG was φ 3 mm (1/e^2^), it can be concluded that the property change in the ICO:H film owing to FHG irradiation did not occur across the entire FHG beam diameter; only the limited area around the laser peak spot could be crystallized. The energy threshold for the property changes of the ICO:H film induced via FHG irradiation, calculated from the above beam diameter and simulated 2D Gaussian curve, was 0.12 μJ.

Subsequently, a line scan of the FHG irradiation was performed at intervals of 0.5 mm and repeated four times. The resulting images of the intensity changes and transmittance spectra are shown in Figure 7g,h, respectively. By expanding the irradiation area in Figure 7e,f, an increase in transmittance was observed over a wider area. The most significant transmittance change occurred in the region spanning ±5 mm of the Z-direction from the centre of the substrate, with a value of 8.1%. This value closely aligns with the 10% transmittance change observed before and after ELI, as shown in Figure 6a. This demonstrates that the optical property changes caused by FHG irradiation are comparable to those caused by ELI.

#### 3.2.2. UV/NIR Camera System

The previous subsection demonstrated the visualisation of the optical property changes in the ICO:H films through FHG irradiation using a single photodetector. However, this method is time-consuming, because it requires multiple movements of the sample to acquire images. Scaling up to larger areas for observation is also challenging, and accurately aligning the FHG and SC light sources at the same spot on the sample surface is not straightforward.

To address these limitations, a real-time imaging system was constructed to observe the optical property changes in the ICO:H films induced by FHG irradiation using UV (395 nm) and NIR (1600 nm) LEDs as light sources and two cameras as detectors, as shown in Figure 2b. To test the system, we used an ICO:H/glass sample that was initially irradiated with FHG under conditions programmed for the XZ stages, as illustrated in Figure 8a, with a rat-shaped configuration, which was confirmed by acquiring an image of the transmittance using SC light (Figure 8d), as explained in Section 3.2.1. Using this sample, we present the UV and NIR images of the ICO:H/glass system captured with the developed camera system in Figure 8c,d, respectively. The 3 cm square sample substrate was affixed to the XZ stage using a metal plate. Both the UV image (Figure 8c) and NIR image (Figure 8d) depict the rat-shaped pattern that can be observed in Figure 8b. The NIR image exhibited a sharper contrast for the pattern than the UV image. This difference was consistent with the transmittance changes before and after laser irradiation in the UV and IR regions, as shown in Figure 6a.

After confirming the distinguishability of light transmittance changes in the ICO:H films induced via FHG irradiation through UV and NIR images, we performed real-time imaging measurements of these variations. Figure 9 shows the temporal UV and NIR images captured during the FHG irradiation of the ICO:H/glass sample (the movie is shown in the Appendix A). Under the FHG irradiation, the profile of the laser light was observed in the UV image, whereas it was absent in the NIR image. When the sample was moved using the XZ stage to trace a square pattern with laser light, the contrast in the NIR image decreased linearly, whereas in the UV image, the FHG light was too intense and impeded the observation of contrast changes. The contrast variations observed in the NIR images are distinctly visible in the differential images. Although the changes in the properties of the ICO:H film induced by the laser light were not discernible in the UV images, they were observable in the NIR images.

## 4. Conclusions

This study was dedicated to the development of real-time monitoring methodologies for the functionalization of TCO films using laser light irradiation. Monitoring techniques have the potential to yield significant advantages, not only in terms of reducing the temporal expenses associated with identifying optimal irradiation conditions but also in facilitating a comprehensive exploration of the underlying mechanisms that govern functionalization phenomena. In this study, an electrical measurement system was firstly devised to track alterations in the electrical properties of ITO/glass samples induced by FHG irradiation. The observed enhancement in resistivity was substantiated with Hall effect measurements and XRD analysis. Nonetheless, the sensitivity of this technique to electrode contact renders it less suitable for application to plastic substrates. The deformation caused by the FHG irradiation of the substrates introduces instability in the context of constant-current measurements.

To address these limitations, alternative optical methodologies, capable of the non-distractive and remote detection of changes in the physical properties of TCO films, have been proposed. Notably, we have shown the efficacy of an NIR imaging technique, enabled by an InGaAs NIR camera, in capturing real-time changes in photophysical properties induced by FHG irradiation. The viability of this technique has been proven. However, the application of a UV camera is hindered by the intense FHG beam, which obscures the irradiated spot within the image. Collectively, the results of these investigations support the proposition of a non-contact and non-destructive monitoring approach employing an InGaAs NIR camera. This approach holds promise for the application of monitoring strategies in the laser-induced functionalization of TCO films on a larger scale via ELI. Since FHG of the Nd:YAG laser that is presently used can functionalize TCO films in a very limited region, and also degrade PET films easily, we are currently planning to apply our optical monitoring system for ELI on TCO films on plastic substrates to establish the correlation between the optical images and functionalization of those samples, which will be reported elsewhere.

## Figures and Tables

**Figure 1 nanomaterials-13-02706-f001:**
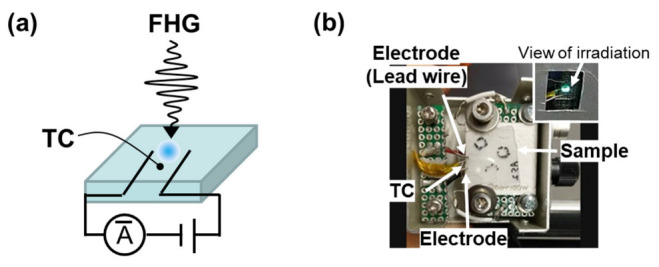
Scheme of electrical measurement setup during FHG irradiation (**a**). Photograph of the measurement conditions (**b**).

**Figure 2 nanomaterials-13-02706-f002:**
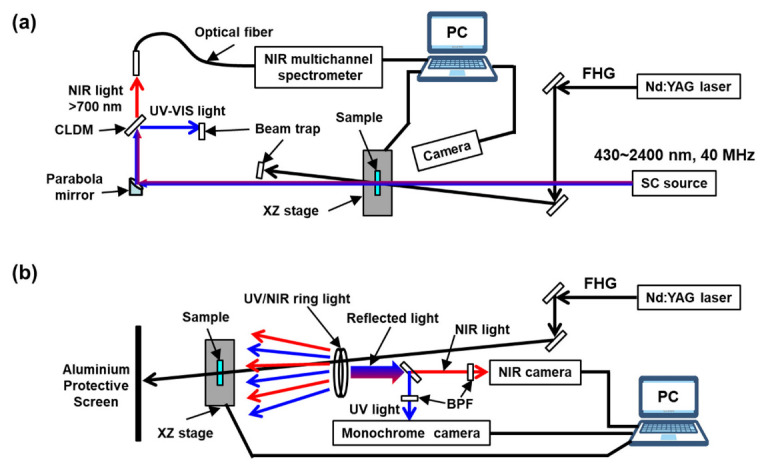
Scheme of the two types of optical measurement setups: (**a**) spectroscopic and single-point scanning measurement setup and (**a**) light reflectance image monitoring setup with two cameras (**b**).

**Figure 3 nanomaterials-13-02706-f003:**
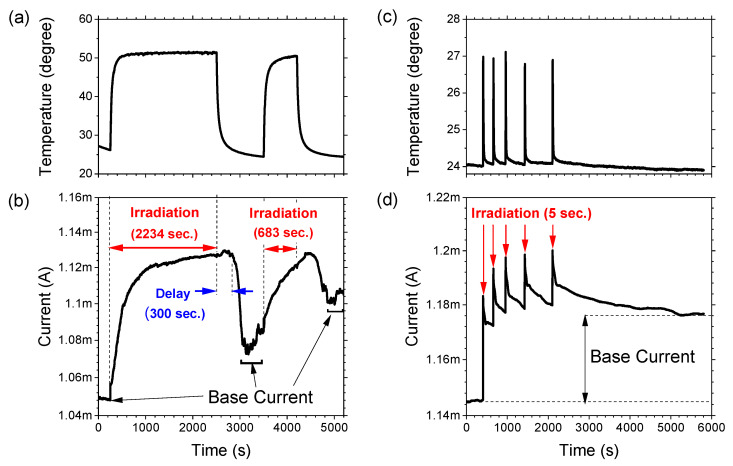
Temporal results of the temperature (**top**) and current (**bottom**) of the ITO film under FHG irradiation. (**a**,**b**) The results for long-term irradiation. (**c**,**d**) The results for short-term irradiation.

**Figure 4 nanomaterials-13-02706-f004:**
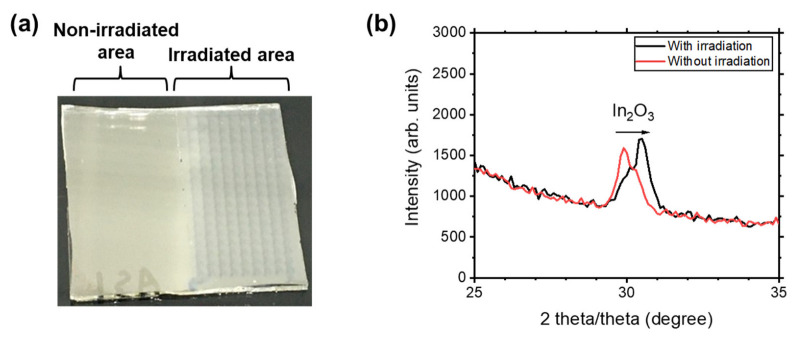
FHG irradiation effect on the ITO film (**a**). Out-of-plane XRD results before and after the FHG irradiation (**b**).

**Figure 5 nanomaterials-13-02706-f005:**
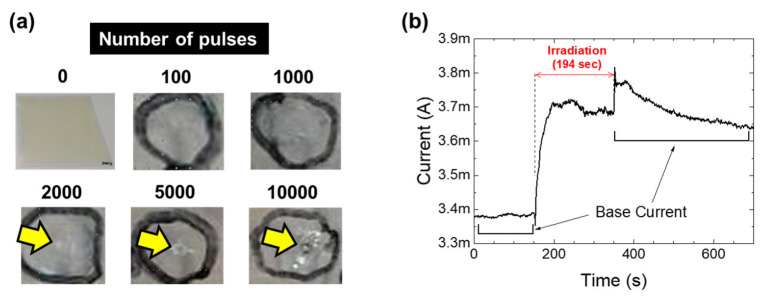
Image of a series of FHG pulses irradiated onto the ICO:H/PET film (**a**). The “0” picture is a full picture of the sample, and the others are enlarged ones. FHG was irradiated inside the black circles. Temporal result of the current of the ICO:H/SiO_2_/PET film under FHG irradiation (**b**).

**Figure 6 nanomaterials-13-02706-f006:**
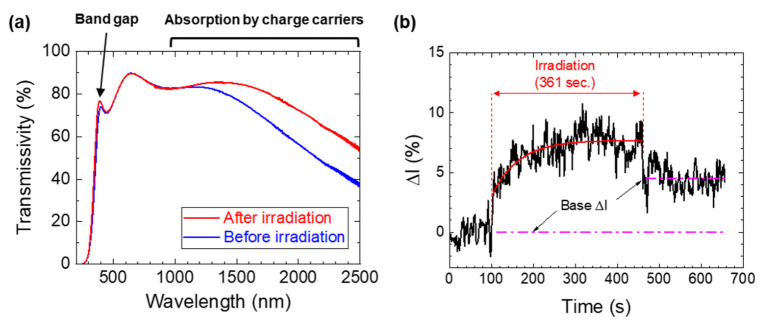
Transmission spectra of the ICO:H/glass system: before (blue) and after (red) ELI (**a**). Temporal results of ΔI of the ICO:H/glass system with FHG irradiation (**b**). The red curve in (**b**) is a simulation curve using an exponential decay function.

**Figure 7 nanomaterials-13-02706-f007:**
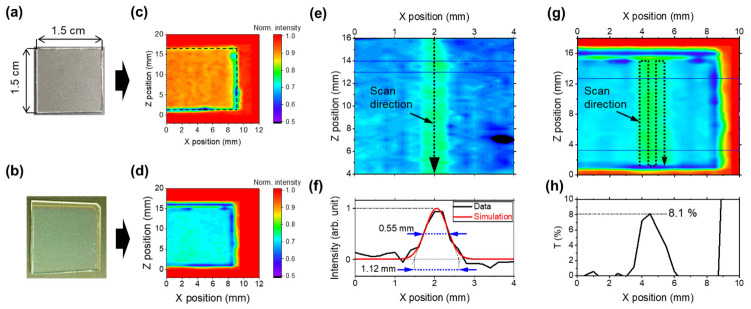
Photo and 2D image of transmission intensity at a wavelength of 1650 nm for SC light: bare glass substrate (**a**,**c**) and ICO:H/glass sample (**b**,**d**). Two-dimensional image after one line scan of the FHG irradiation (**e**) and its intensity profile (**f**). Two-dimensional image after four line scans of the FHG irradiation (**g**) and its transmittance profile (**h**).

**Figure 8 nanomaterials-13-02706-f008:**
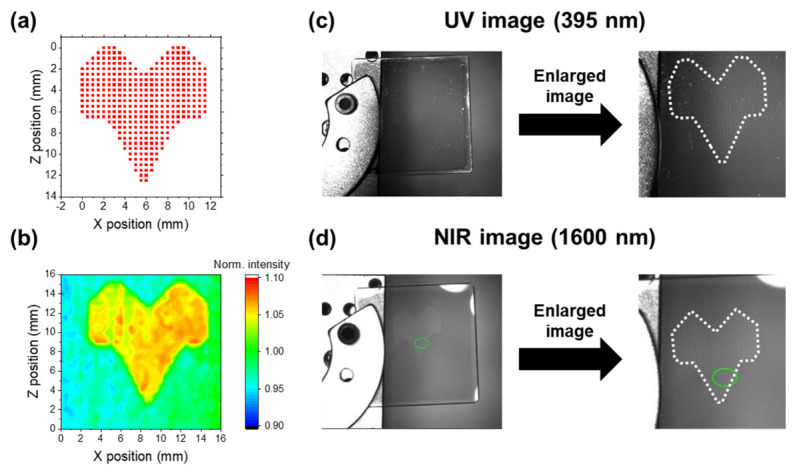
Simulation of FHG irradiation (**a**) and its result for the ICO:H/glass (**b**) for an SC light source. UV (**c**) and NIR (**d**) images of the sample with the irradiated pattern in (**b**). The left and right images are full pictures and enlarged images of the irradiated position (blue broken pattern), respectively. For this system, the intensity accumulated in a given region—for instance, the green circle region in (**d**)—can be monitored in real time.

**Figure 9 nanomaterials-13-02706-f009:**
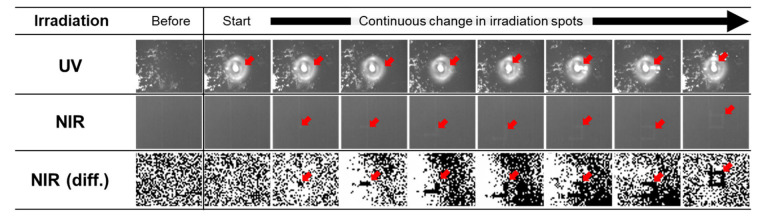
Temporal UV and NIR images of FHG irradiation on the ICO:H/glass samples. The images are selected from a movie shown in Appendix A. The bottom images are differentials of the NIR images. The red arrows indicate the location of the FHG light.

**Table 1 nanomaterials-13-02706-t001:** Electrical properties of the ITO films obtained via Hall effect measurements.

Sample Area	Resistivity (Ωcm)	Carrier Concentration (cm^−3^)	Hall Mobility (cm^2^/V s)
Non-irradiated	5.25 × 10^−4^	5.48 × 10^20^	21.3
Irradiated	2.73 × 10^−4^	1.09 × 10^21^	20.9

**Table 2 nanomaterials-13-02706-t002:** Repetition rate dependence on the resistance and temperature increase for ITO films.

Repetition Rate (Hz)	Resistance (Ω)		DT (°C)
	Before Irradiation	After Irradiation	
1	184 ± 10	167 ± 9	Not detectable
10	183 ± 14	168 ± 12	~0.1 ± 0.005
100	185 ± 16	163 ± 14	0.3 ± 0.028
1000	186 ± 6	166 ± 6	37.7 ± 0.005

## Data Availability

The data supporting the findings of this study are available upon request from the authors.

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
