# Peer review of "The Real-Time Monitoring of the Laser-Induced Functionalization of Transparent Conductive Oxide Films"

_nanomaterials, 2023, doi:10.3390/nano13192706_

Round 1

Reviewer 1 Report

Comments:

This work reports electrical and optical characterizations of crystallization change during laser irradiation for sputtered oxide films, which is meaningful for practical applications requiring high conductivity, especially for flexible electrodes. Experiments are well designed and exhibited. I recommend it to be published after major revisions listed below are made.

1.     The main issue is the lack of quantitative relationship between optical change and crystallization degree of irradiated films, which will be the guide for irradiation condition optimization. As discussed in the former part on electrical measurements, irradiation results could be reflected by current or temperature variations considering the carrier concentration, so irradiation conditions were optimized. However, for flexible electrodes, no direct connection of optical properties with film crystallization degree was given. Optical change with electrical parameters or XRD peak widths (FWHM) should be established.

2.     Pay attention to the expression clarity. Meanings of “However, for high-efficiency solar cell applications, solid-phase crystallised hydrogen (H)-doped In2O3 (IO:H) and H and transition metals (such as Ce and ICO:H), which exhibit high transparency over a wide range from visible to near-infrared (NIR), have been frequently investigated” on P1 and “Although the irradiation energy density of our Nd:YAG laser was lower by a factor of 2.5, compared with the excimer laser used for ELI in our group, laser-induced functionalisation could be expected owing to the much higher peak power of the Nd:YAG laser by a factor of 8.6, as we calculated before experiments” on P2 are unclear.

3.     English writing and format should be improved, such as “Therefore. to facilitate the exploration of the laser irradiation conditions of TCO films” on P2. Introduction should be reorganized with emphasis on the principle and effectiveness of NIR image variation during ELI considering the carrier concentration, while technique of time-resolved TEM which was not involved in the manuscript, can be less talked about. Moreover, the manuscript researches only on crystallization of sputtered films, so the word “Functionalisation” in the title is advised to be replaced by “Crystallization”.

Meanings of “However, for high-efficiency solar cell applications, solid-phase crystallised hydrogen (H)-doped In2O3 (IO:H) and H and transition metals (such as Ce and ICO:H), which exhibit high transparency over a wide range from visible to near-infrared (NIR), have been frequently investigated” on P1 and “Although the irradiation energy density of our Nd:YAG laser was lower by a factor of 2.5, compared with the excimer laser used for ELI in our group, laser-induced functionalisation could be expected owing to the much higher peak power of the Nd:YAG laser by a factor of 8.6, as we calculated before experiments” on P2 are unclear.

Reviewer 2 Report

In this work, the authors a development of real-time monitoring methodologies for the functionalisation of TCO films using laser light irradiation was shown. An electrical measurement system was devised to track alterations in the electrical properties of ITO/glass samples induced by FHG irradiation effectively. The authors the efficacy of an NIR imaging technique enabled by an NIR camera in capturing real-time changes in photophysical properties induced by FHG irradiation was shown.

Contents are interesting, and I can recommend a publication after taking into account below comments.

1. Table 2: Resistance and temperature - It is necessary to specify the measurement error / accuracy.

2. Figure 7 c, d, Figure 8 b: Add numerical values to the color legend.

-

Reviewer 3 Report

This work demonstrates that laser treatment of transparent conducting oxide (TCO) films with picosecond pulses of the fourth harmonic of a neodymium laser at a wavelength of 266 nm makes it possible to reduce the resistivity of the films. Electrical and non-contact optical methods for real-time monitoring of laser modification of TCO films have been proposed and developed. Overall, the work makes a good impression. The text of the article is quite well edited, although there are some typos, for example, an extra period in line 84 after the word “therefore”. The figures and their captions well illustrate the results obtained.

However, the content of the article and the main results obtained related to the optical diagnostics of the laser modification process of TCO films are not very suitable for the subject of the journal Nanomaterials. In my opinion, the article can be successfully published in the journal Applied Sciences.

English language fine. No issues detected.

Reviewer 4 Report

The authors developed a real-time electronic and two real-time optical monitoring setups to investigate the effects of Laser-induced functionalization on transparent conductive oxide (TCO) films, and compared the functions between the setups. The electric-mode setup can record the current and the temperature of TCO films in real time, and the two optical setups can record the transmittance changes and provide the single-spot or 2D-dimensional images, depending on the design of the setup. The setup designs, to some extent, open the mind for real-time monitoring systems which potentially can be used for investigations of TCO parameters under laser radiation.

There is an ambiguity about the deformation of studied PET flexible TCO films caused by electric mode. The authors attributed to its electric contact. If so, why did the authors not use optical mode to study the PET TCO films since the two optical setups do have electric contact?

Some typos and grammars need to be checked.

 1.     Line 31-35. There are two “However” used, which make the phrases not fluent.

 2.     Line 83-84. After “Therefore”, it should be comma not period.

Round 2

Reviewer 1 Report

Acceptable in current form.